# Unlike Brief Inhibition of Microglia Proliferation after Spinal Cord Injury, Long-Term Treatment Does Not Improve Motor Recovery

**DOI:** 10.3390/brainsci11121643

**Published:** 2021-12-13

**Authors:** Gaëtan Poulen, Sylvain Bartolami, Harun N. Noristani, Florence E. Perrin, Yannick N. Gerber

**Affiliations:** 1MMDN, Univ. Montpellier, EPHE, INSERM, Montpellier, France; g-poulen@chu-montpellier.fr (G.P.); Sylvain.Bartolami@umontpellier.fr (S.B.); harun.noristani@ibn-lab.com (H.N.N.); florence.perrin@umontpellier.fr (F.E.P.); 2Department of Neurosurgery, Univ. Montpellier, CHU, Montpellier, France; 3Institut Universitaire de France (IUF), France

**Keywords:** spinal cord injury, reduced microglia proliferation, colony stimulating factor 1 receptor, GW2580, treatment-duration

## Abstract

Microglia are major players in scar formation after an injury to the spinal cord. Microglia proliferation, differentiation, and survival are regulated by the colony-stimulating factor 1 (CSF1). Complete microglia elimination using CSF1 receptor (CSF1R) inhibitors worsens motor function recovery after spinal injury (SCI). Conversely, a 1-week oral treatment with GW2580, a CSF1R inhibitor that only inhibits microglia proliferation, promotes motor recovery. Here, we investigate whether prolonged GW2580 treatment further increases beneficial effects on locomotion after SCI. We thus assessed the effect of a 6-week GW2580 oral treatment after lateral hemisection of the spinal cord on functional recovery and its outcome on tissue and cellular responses in adult mice. Long-term depletion of microglia proliferation after SCI failed to improve motor recovery and had no effect on tissue reorganization, as revealed by ex vivo diffusion-weighted magnetic resonance imaging. Six weeks after SCI, GW2580 treatment decreased microglial reactivity and increased astrocytic reactivity. We thus demonstrate that increasing the duration of GW2580 treatment is not beneficial for motor recovery after SCI.

## 1. Introduction

Traumatic spinal cord injury (SCI) has a worldwide incidence of 10.5 cases per 100,000, with 600,000 to 900,000 additional cases annually [1]. SCI leads to heterogeneous deficits that depend on the location of the injury and lesion severity. There are currently no curative treatments on any deficits induced by SCI. After SCI, the primary tissue damages are characterized by disruption of the blood–spinal cord barrier, cell death, myelin degeneration, infiltration of immune cells, and axon degeneration [2,3]. Thereafter, worsening of the functional neurological outcome results from a cascade of secondary events, including the production of pro-inflammatory factors and reactive oxygen species, as well as ischemia and edema [4].

The glial scar that forms after SCI constitutes a physical and chemical barrier that prevents spontaneous axonal regeneration and recovery of neurological functions. The glial scar is mostly composed of glial cells, including microglia, whose responses occur within minutes following SCI. Early after injury, microglia display an expression profile characteristic of cell proliferation [5]. Moreover, an extensive microglial proliferation persists over 2 weeks post-injury [6].

Therapeutic strategies targeting a microglia response are attractive approaches to promote functional regeneration after injury. In particular, the macrophage colony stimulating factor (CSF1) that participates in proliferation, differentiation, and survival of myeloid lineage cells had been targeted using pharmacological approaches. PLX5622 and PLX3397 are CSF1 receptor (CSF1R) inhibitors that eliminate all microglial populations. These inhibitors had been previously tested in SCI. Mice with a thoracic (T10/T11) spinal cord contusion that were continuously treated with PLX5622 between 3 weeks prior to SCI to 5 weeks post-injury displayed worsening motor recovery [6]. A six-week PLX5622-treatment after a T10 spinal cord contusion enhanced cognitive behavior but failed to improve general motor function [7]. Similarly, eradication of microglia using an oral PLX3397-treatment in mice between 7 days before SCI to 4 weeks after severe T10 spinal cord crush led to impaired locomotor function and exacerbated tissue damage [8].

Conversely, we have demonstrated a beneficial effect of a post-SCI oral GW2580 treatment. GW2580 is another inhibitor of CSF1R that specifically inhibits microglia proliferation [9]. A GW2580 treatment beginning 4 weeks before the lesion and continuing for 6 weeks after SCI was associated with an improvement in fine motor recovery and a decrease in gliosis after a T9 spinal cord hemisection [10]. With a translational objective, we further extended our approach to a transient post-SCI GW2580 treatment in mice and nonhuman primates [11]. One-week (in mice) and two-week (in nonhuman primates) post-injury GW2580 treatment improved functional motor recovery and was associated with a modulation of tissue damage. Strikingly, nonhuman primates displayed greater improvement in functional recovery than mice. In mice, a short-term inhibition of microglia proliferation led to a decrease in microgliosis 2 weeks after the lesion but an increase in IBA1 reactivity 6 weeks after SCI. Of great interest, microglial reactivity returned to baseline 3 months after SCI in nonhuman primates [11].

Taken together, in the context of SCI, these studies highlight the importance of a selective inhibition of microglia proliferation rather than a complete eradication of all microglia. One remaining question is whether prolonged inhibition of microglia proliferation after SCI would further increase GW2580-induced beneficial effects. Therefore, we assessed the effect of a 6-week GW2580 oral treatment after hemisection in mice on functional recovery and tissue reorganization. We demonstrate that prolonged GW2580 treatment is not beneficial after SCI.

## 2. Materials and Methods

### 2.1. Study Approval

Studies were approved by the local ethic committee (n°36), by the Veterinary Services Department of Hérault, and by the French Ministry of National Education Higher Education and Research (authorization n°34118). Experiments followed the European legislative, administrative and statutory measures (EU/Directive/2010/63), and the Animal Research: Reporting of In Vivo Experiments (ARRIVE) guidelines. All efforts were made to reduce the number of animals and their suffering.

### 2.2. Spinal Cord Injury

Heterozygote transgenic female mice expressing enhanced green fluorescent protein (eGFP) in central nervous system (CNS) resident microglia and circulating peripheral monocytes (CX3CR1^+/eGFP^) were used in all experiments (obtained from Dr. Dan Littman, Howard Hughes Medical Institute, Skirball Institute, NYU Medical Center, New York, NY, USA). CX3CR1^+/eGFP^ mice express eGFP downstream of the *Cx3cr1* promoter. Transgenic line was maintained on a C57BL/6 background (The Jackson Laboratory, Bar Harbor, ME, USA; [12]). Mice were housed in controlled hygrometry and temperature and were subjected to a 12 h light/dark cycle with access to food and water ad libitum. At three months of age, mice were anesthetized with isoflurane gas (induction at a flow rate of 3% maintained to 1.5% during surgical procedure). First, we performed a vertebral thoracic 9 level (T9) laminectomy, followed by a lateral hemisection of the spinal cord (HS) performed under a microscope using a micro knife (10315–12, fine science tools (FST)), as described previously [13,14]. Muscles and skin were sutured, and animals remained under visual monitoring for a 2 h recovery period immediately after surgery. Mice were then placed back in their home cages. Bladders were emptied manually twice a day until recovery of bladder sphincter control. Bodyweights were monitored prior to surgery and then daily throughout 6 weeks of the study. Following surgery, mice were randomly assigned to untreated or GW2580-treated groups. Number of mice: 32 hemisected (16 untreated and 16 GW2580-treated). Owing to unexpected outcomes, such as paraplegia or scoliosis, 4 mice were excluded from the study after surgery (2 in each group).

### 2.3. GW2580 Oral Treatment

Oral treatment started immediately after SCI and ended 6 weeks post-lesion when the last behavioral analysis protocol was completed (Figure 1A). Mice were fed with a regular chow (A04, maintenance diet, SAFE diet, Augy, France) or with the same diet containing 0.1% GW2580 (LC Laboratories, Woburn, MA, USA). GW2580 food pellets were prepared as previously described [10,11]. Food intake was monitored daily throughout the study duration.

### 2.4. Behavior

Motor behavior was assessed at 1 week and 1 day prior to injury, followed by acquisitions at 3 days, 5 days, and then once a week up to 6 weeks after injury (Figure 1A). Habituation to the experimenter, novel environment, and apparatus was done to limit bias due to stress. Mice were placed on the CatWalk^®^ glass plate as well as in the open field arena for 10 min sessions without recording 7 days prior to the first motor assessment. An additional group of 3 uninjured mice were assessed similarly for 6 weeks.

Gait analysis was done using CatWalk^®^ (Noldus, Wageningen, The Netherlands), and data were analyzed with CatMerge^®^ software (InnovationNet, Tiranges, France), as previously described [10,14,15,16,17]. We used CatWalk^®^ version 7.1.9 for this study. Briefly, 6 runs per animal were recorded for each time-point with a minimum inclusion criterion of at least 3-full step sequence patterns per run and a minimum of 3 runs crossed at similar speed. Several parameters were quantified including regularity index, base of support, and the ‘‘maximum contact at %’’ that represents the time of maximum contact relative to stand. Ipsilateral hind-paw were difficult to detect using CatWalk^®^ between day 3 to day 14 post-injury, therefore, representative graph bars obtained with CatWalk^®^ only show values from 2 to 6 weeks after SCI (Figure 1B–D). Spontaneous motor activity was monitored using the open field test. Mice were placed in an empty test arena (50 × 50 cm), and their spontaneous locomotor activity was recorded by a camera located over the arena. Recording sessions lasted 8 min. Analyses were done using a video tracking software (EthoTrack^®^, InnovationNet, Tiranges, France). We measured the total distance covered per mice during each recording session.

### 2.5. Ex Vivo Diffusion MRI (DW-MRI)

Six weeks after SCI, animals were injected with a lethal dose of tribromoethanol (i.p., 500 mg/kg, Sigma-Aldrich, Darmstadt, Germany). Mice were intracardially perfused with phosphate saline buffer (PBS, 0.1 M, pH 7.2), followed by 4% paraformaldehyde (PFA, pH 7.2, Sigma Aldrich, Darmstadt, Germany) in 0.1 M PBS. Spinal cords were placed for 2 h in 4% PFA and then stored in 1% PFA until DW-MRI acquisition. Six mice per group (untreated vs. GW2580-treated mice) were randomly selected for ex vivo MRI recordings. DW-MRI were acquired and analyzed as described previously [17]. In brief, samples were placed in a custom-made solenoid coil constructed for spinal cord acquisition [18] and introduced in a 9.4 T apparatus (Agilent Varian 9.4/160/ASR, Santa Clara, CA, USA). DW-MRI parameters: delta = 6.88 ms, G = 10 G/cm^−1^, separation = 15.05 ms, TR = 1.580 ms, TE = 30.55 ms, AVG = 30, FOV=10 mm × 10 mm, slices = 36, thickness = 1 mm without gap, and acquisition matrix = 128 × 128. Segmentations were done using Myrian software (Intrasense, Montpellier, France) to quantify the lesion area (% of the total axial spinal surface), the extension of the lesion along the rostro–caudal axis, and the volume of the lesion (corresponding to the area under the curve). The longitudinal apparent diffusion coefficient (LADC) was measured on a 0.8 cm segment rostral and caudal to the lesion, excluding the lesion site slice. After MRI acquisitions, spinal cords were immersed in 30% sucrose diluted with 0.1 M PBS for cryoprotection, embedded with Tissue-Tek (Sakura, Alphenaanden Rijn, The Netherlands), frozen, and kept at −20 °C until processing.

### 2.6. Histology

Axial spinal cord cryosections (14 µm) were collected on Superfrost Plus slides using a cryostat (Microm HM550, Thermofisher Scientific, Waltham, MA, USA). Cryosections were washed with 0.1 M PBS then placed for 30 min in 1% hydrogen peroxide diluted in 0.1 M PBS. Unspecific binding sites were blocked for 2 h with 1% BSA and 0.1% Triton X-100 in 0.1 M PBS. Axial sections were then incubated for 48 h at 4 °C with the primary antibody. We used either rabbit anti-IBA1 ((ionized calcium binding adaptor molecule 1), Wako Pure Chemical Industries, Osaka, Japan; 1:1000) or rabbit anti-GFAP ((glial fibrillary acidic protein), Dako, Glostrup, Denmark; 1:1000) antibodies. After several PBS washes, sections were incubated in a 1:500 dilution of anti-rabbit HRP-conjugated secondary antibody (Jackson Immunoresearch, Stratech Scientific Ltd., Soham, UK) for 2 h at room temperature. Sections were rinsed in TRIS 0.1 M, and the peroxidase reaction product was revealed using a DAB peroxidase substrate kit (Vector Labs, Burlingame, CA, USA). The reaction was stopped by rinsing the sections with TRIS 0.1 M. Section dehydration was done using ascending ethanol concentration (70, 80, 95%, and 100%) and cleared with xylene. Coverslips were applied using Eukitt mounting media (Sigma Aldrich, Gilligham, UK), and slides were dried for 48 h before imaging. Immunohistochemistry for all animals were done simultaneously for a given target epitope. All reagents for histology and immuno-histology (if not specified otherwise) were from Sigma Aldrich, Gilligham, UK. Eight mice per group (untreated vs. GW2580-treated mice) were randomly selected for IBA1 and GFAP histological analysis.

### 2.7. Imaging and Quantifications

All slides were scanned using NanoZoomer^®^ (NanoZoomer Digital Pathology System and NDP view software, Hamamatsu City, Japan). This apparatus uses constant exposure time as well as light intensity. Micrographs were exported at a constant magnification using NDPview^®^ software. Images were exported using the same parameters for a given staining. All quantifications were done blindly. Optical density (OD) was measured at different distances from the lesion site and at the epicenter using ImageJ software (National Institutes of Health, Bethesda, MD, USA), as described previously [14,19,20]. OD quantifications included the grey matter, the white matter (excluding the *dorsal funiculus*), and the *dorsal funiculus* that is a notable zone presenting high rate of microglia/monocytes activation after SCI [14,19]. Optic densities (IBA1 and GFAP) were measured on a 0.63 cm segment rostro–caudal to the lesion in serial axial sections of the spinal cord (630 µm interval between each section). Due to section detachment from the slides, 1 mouse (GW2580-treated) was excluded from the IBA1 OD analysis, and 3 mice (2 untreated and 1 GW2580 treated) were excluded from the GFAP OD analysis.

### 2.8. Statistics

Two-way repeated-measure analysis of variance (ANOVA) with Bonferroni post-hoc tests were used for Catwalk^®^ and open-field analysis. Unpaired *t*-tests were used for longitudinal diffusivity comparison and analysis of OD measurements in histology. Significance was accepted at *p* ≤ 0.05. Data were analyzed using GraphPad Prism 5.0 (GraphPad Software Inc., San Diego, CA, USA) and expressed as the mean ± standard deviation of the mean (SEM).

## 3. Results

### 3.1. Long-Term GW2580 Treatment after SCI Does Not Improve Motor Recovery

Recently, we have shown that transient (1 week) inhibition of microglia proliferation improves motor recovery after SCI [11]. To investigate whether prolonged inhibition of microglia proliferation further improves motor outcomes, we studied the effect of long-term (6 weeks) GW2580 treatment starting immediately after SCI (Figure 1A). We first compared motor function of uninjured vs. injured mice to verify that T9 lateral hemisection of the spinal cord induced a significant and quantifiable impact on motor activity using CatWalk (Appendix A) and open field (Appendix A). We then assessed motor recovery of hemisected mice (*n* = 28) over 6 weeks post-injury. Fifty percent of the mice were treated per os with GW2580 starting immediately after SCI and continued throughout the 6 weeks post-SCI, whilst the other 50% were untreated. Using CatWalk^®^ (Figure 1A), we showed no difference between the GW2580-treated mice and the untreated controls in several static and dynamic parameters, including the “regularity index” (Figure 1B), the “base of support” (Figure 1C), and the “max contact” (Figure 1D). In addition, the overall distance covered in 8 min in an open field (Figure 1A,E) was similar between the two groups. These data suggest that sustained inhibition of microglia proliferation using GW2580 has no effect on motor recovery in mice.

### 3.2. Long-Term GW2580 Treatment after SCI Has No Effect on Tissue Reorganization

We next used ex vivo diffusion-weighted magnetic resonance imaging (DW-MRI) to analyze the effect of long-term GW2580 treatment on the lesion volume and the spinal cord microstructure. Using DW-MRI, we could clearly discriminate the lesion epicenter (Figure 2A, underlined in red) as well as the spared white and grey matters. Six weeks after SCI, the percentage of damaged tissue at the lesion epicenter, the rostro–caudal lesion extension, and the lesion volume (area under the curve 121.1+/−9.35 and 117.5+/−9.45 in the untreated and GW2580 treated mice, respectively) were similar in both groups (Figure 2B). Using luxol fast blue and toluidine blue staining, we further confirmed the absence of difference in lesion size between the two groups (data not shown). Then, we quantified longitudinal (Figure 2C,D) and transverse diffusivities on a 1.6 cm-spinal cord segment centered on the lesion site separately in the white matter (WM), excluding the *dorsal funiculus* (DF) and the DF. No differences were detected between groups in the longitudinal apparent diffusion coefficient (LADC) both rostral (Figure 2C) and caudal (Figure 2D) to the lesion.

The transversal apparent diffusion coefficient (TADC) was also similar in the WM and the DF of GW2580-treated as compared to untreated animals (data not shown). Taken together, these data suggest that a long-term GW2580 treatment post-SCI has no effect on the secondary lesion extension and tissue reorganization.

### 3.3. IBA1-Microglial Expression Is Decreased 6 Weeks after SCI in the GW2580-Treated Mice

To investigate the effect of prolonged inhibition of microglia proliferation, we quantified microglial reactivity at 6 weeks after SCI on a 1.3 cm-perilesional segment of the spinal cord (Figure 3A–L). We observed a significant decrease of IBA1 expression in the GW2580-treated group compared to the untreated group at the lesion site (GW2580-treated: 29.99 ± 1.916; untreated: 37.89 ± 1.276; *p*-value = 0.0038) (Figure 3C). Along this line, microglial reactivity was significantly reduced in the WM, ipsilateral and caudal to the lesion, in the treated group in comparison with the control group (GW2580-treated: 19.74 ± 1.650; untreated: 24.05 ± 1.061; *p*-value = 0.0419) (Figure 3I), and in the DF contralateral to the lesion (GW2580-treated: 15.59 ± 0.8517; untreated: 19.14 ± 0.6393; *p*-value = 0.0049) (Figure 3L). By contrast, no differences between groups were observed in the rostral segment (Figure 3E,H,K). Caudal to the lesion, the grey matter (GM) (Figure 3F), the contralateral WM (Figure 3I), and the ipsilateral DF (Figure 3L) sides also showed similar optic density values between untreated and GW2580-treated groups. These data suggest that, in mice, a long-term GW2580 treatment induces a persistent decrease of IBA1 expression 6 weeks after SCI within the lesion site and caudal to the lesion.

### 3.4. Astrocytic Reactivity Is Increased 6 Weeks after SCI with a Long-Term GW2580 Treatment

We next quantified astrocytic reactivity at 6 weeks after SCI on a 1.3 cm-perilesional segment of the spinal cord using GFAP (Figure 4A–L). We found a significant increase of GFAP expression on the contralateral side caudal to the lesion in both grey (GW2580-treated: 31.20 ± 2.298; untreated: 23.15 ± 2.314; *p*-value = 0.0320) (Figure 4F) and white (GW2580-treated: 28.76 ± 1.500; untreated: 22.93 ± 1.906; *p*-value = 0.0328) (Figure 4I) matters in the treated group in comparison with the control group. Contrariwise, astrocytic reactivity was similar in the two groups at the lesion site (Figure 4C), as well as rostral to the lesion in the GM (Figure 4E) and in the WM (Figure 4H). DF also showed no difference in GFAP reactivity between the two groups (Figure 4K,L). These data demonstrate that an increased astrocytic reactivity is induced caudal to the lesion upon prolonged GW2580 treatment 6 weeks after SCI.

## 4. Discussion

Here, we show that prolonged GW2580 treatment starting immediately after SCI has no impact on functional recovery. Contrariwise, we have previously shown that a continuous GW2580 treatment starting between 4 weeks prior to SCI and 6 weeks post-injury, (Figure 5B) improved fine motor outcomes in mice [10] compared to untreated animals (Figure 5A). Additionally, sustained inhibition of microglia proliferation pre- and post-SCI triggered reduced microgliosis 6 weeks after SCI (corresponding to the end of the treatment, Figure 5B) that may indicate long-term persistence of microglial proliferation after injury. GW2580 delivery prior to SCI is effective in inhibiting early microglia proliferation (just after SCI), converse to an oral-treatment initiated after the injury. Indeed, mice reduce their food intake at least for over 24 h following SCI. In a second study, we have shown that a short-term inhibition of microglia proliferation through GW2580 per os delivered during a 1-week post-SCI (Figure 5C) led to a better motor recovery and preservation of tissue structure [11]. Thus, post-lesion GW2580-induced improvements may reflect that microglial proliferation does not occur within the first few hours after injury. A previous report showed that the peak in microglia proliferation occurs at 1 week following T10/T11 spinal cord contusion in mice [6]. Following a 1-week GW2580 treatment starting immediately after SCI in mice, we observed an increased microgliosis six weeks post-lesion (i.e., five weeks after completion of the treatment, Figure 5C) [11]. It may represent either a compensatory response to the inhibition of microglia proliferation or a persistence of microglial proliferation long after injury. In the current study, we confirm that at the end of a long-term GW2580 treatment after injury, there is a reduced microgliosis (Figure 5D), again, speaking in favor of a persistent microglia proliferation after injury. However, the absence of an improved recovery points towards the importance of treatment duration.

Beneficial effects of GW2580 administration in several animal models of neurodegenerative diseases have been demonstrated with various durations of treatment. GW2580 was given for only a few days in a mouse model of Parkinson’s disease [21], while it was administered over several weeks in a mouse model of amyotrophic lateral sclerosis [22], Alzheimer’s [23], and prion diseases [24]. In all cases, GW2580 treatment had beneficial outcomes on neurological symptoms. After a focal brain injury in adults, an enhanced microglial response associated with a reduced functional recovery was reported for animals previously exposed to early life stress (P14–21, 30 min/day) [25]. Strikingly, GW2580 given concomitantly with early stress exposure diminished the effects induced by a brain injury in adults. Conversely, GW2580 treatment just after an adult brain injury in mice that were previously exposed to early life stress did not improve recovery nor microglial response. Altogether, these data point towards the importance of the time window for GW2580 treatment. To better understand mechanisms that underlie the beneficial effects induced by a short-term GW2580 administration after SCI, we previously investigated the functional state of microglia [5]. Using a cell-specific RNA-Seq analysis, we examined microglial transcriptomic changes between control and GW2580-trated mice at one week after injury [5]. Microglia from GW2580-treated mice displayed a reduced inflammatory response that may explain the better functional recovery. Notably, GW2580 treatment reversed the up-regulation of nine genes induced by SCI that are involved in immune response, cell proliferation, and cell migration [11].

We have previously shown that a short-term GW2580 treatment is sufficient to improve motor recovery after SCI [11]. In the current study, we demonstrate that extending GW2580 treatment post-SCI abolishes its beneficial effects. It most likely reflects that in mice, a 1-week treatment is long enough to cover a peak of detrimental microglial proliferation. Along this line, we have shown that a 2-week GW2580 treatment after SCI in lemurs was beneficial to motor recovery [11]. This is consistent with the delayed inflammatory response reported in nonhuman primates, as compared to rodents [26,27]. Of interest, even if prolonged GW2580 treatment post-SCI is not effective on promoting motor recovery, it is not harmful either, unlike the administration of PLX3397 [8,28] or PLX5622 [6] in the SCI context. However, in mice, a prolonged microglial depletion with PLX5622 at a chronic stage of a traumatic brain injury reduced inflammation and improved neurological function [29]. Similarly, duration of microglia eradication in other diseases led to various outcomes. A 3-week PLX3397-treatment in a mouse model of temporal lobe epilepsy had no effect on memory deficits [30], whereas a 3-month treatment in a mouse model of Alzheimer’s disease reduced amyloid plaque formation [31]. Differences in the functional outcome observed after a complete or selective subset of microglia depletion in neurological diseases and trauma may also reflect a molecular and functional heterogeneity of microglia and their responses, depending on the location in the CNS (for review see [32]), as well as the type of damage.

Importantly, microglia are also characterized by numerous crosstalks with other cells of the CNS; thus, their modulation also modifies responses of neighboring cells [32]. Along this line, treatment with GW2580 during the 6 weeks after SCI not only resulted in a decreased microglial reactivity at the epicenter and caudal to the lesion (Figure 5D), but also an increase in GFAP reactivity caudal to the lesion. Conversely, a 1-week GW2580 treatment led to an increase in microglial reactivity 6 weeks after SCI compared to untreated mice (Figure 5C) [11], although astrocyte reactivity was similar in both groups (personal communication). Continuously inhibiting microglial proliferation beyond the peak of proliferation may interfere with the phagocytose of cellular debris and may therefore participate in the persistence of inhibitory signals associated with the debris and contribute in the absence of improvement in functional motor recovery. The increase in GFAP reactivity may reflect the astrocytic proliferation and/or activation that results from persistent microglial inhibition and may correspond to compensatory and/or detrimental phenomena. Similarly, cross-effects have been shown after complete microglial depletion using PLX3397 in SCI. Indeed, besides a strong disruption of the formation of the glial scar and an increased infiltration of immune cells, a delayed astrocytic proliferation had also been reported [8]. Deleterious effect of astrocytes after SCI have been suggested, in particular, through the expression of chemokines, cytokines, and the nuclear factor Kappa B that are all detrimental to oligodendrocytes and thus to remyelination [33]. In addition, reactive astrocytes are involved in the expression, amongst other proteins, of chondroitin sulfate proteoglycans (CSPGs) that inhibit axonal regeneration after SCI (for review see [34]). Moreover, it had been shown that astrocytes change their phenotype depending on the environment (for review see [35]); thus, a 6-week GW2580 treatment may induce a different response of activated astrocytes than a 1-week treatment duration [11] that eventually leads to an absence of functional recovery. Interestingly, we have shown that microglial expression of *Cspg4* is upregulated (7.89-fold change) by SCI [5] and is subsequently downregulated (−2.74-fold change) by a 1-week GW2580 treatment after lesion [11]. Therefore, an increased GFAP reactivity may also contribute to the absence of improvement in functional recovery that we observed after long-term GW2580 treatment. Deciphering the intercellular bidirectional molecular crosstalk between microglia and astrocytes is at the cutting edges of glial investigations in physiological and pathological conditions, including CNS trauma (for review see [36,37,38]). Taken together, modulation of the microglial response, and to a larger extent, the glial response, requires precaution in the aim of pre-clinical translation since the equilibrium between positive and negative roles of glial cells on axonal regrowth is highly complex.

In conclusion, we demonstrate here that a prolonged inhibition of microglia proliferation during the 6 weeks after SCI has no effect on functional outcome. A comparison with our previous study highlights that a short-term depletion of microglia proliferation is sufficient to improve functional motor recovery [11]. There is, thus, no benefit to extend the GW2580 treatment duration that may be advantageous for transposition to clinics.

## Figures and Tables

**Figure 1 brainsci-11-01643-f001:**
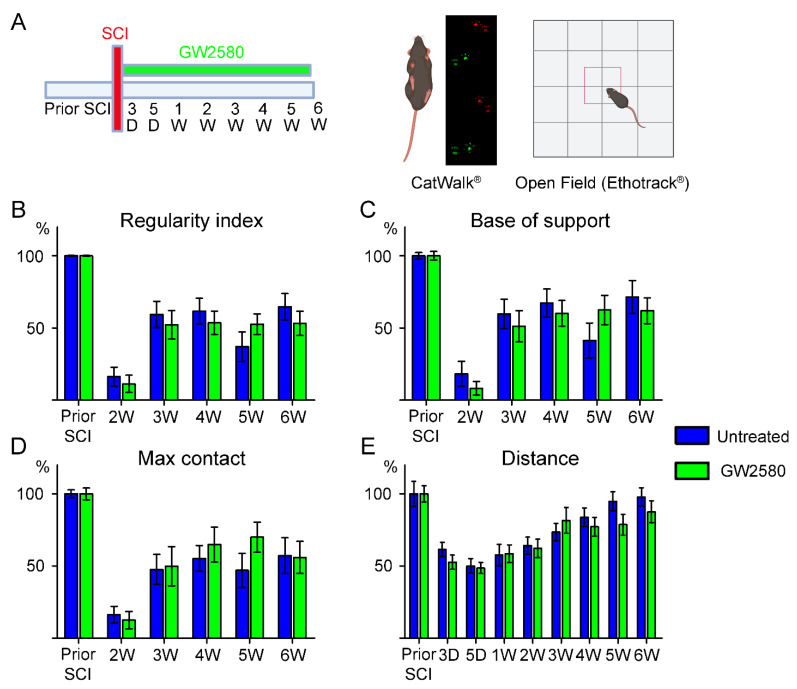
Effects of a long-term GW2580 treatment post-injury on motor activity. Timeline of the experimental procedure corresponding to 6 weeks of oral GW2580 treatment (or control food) after injury and longitudinal behavioral follow-up using a combination of gait and open field analyses (**A**). Parameters used to analyze fine motor activity using Catwalk^®^ include: regularity index (**B**), hindlimbs base of support (**C**), and max contact (%) of the ipsilateral hind paw (**D**). Analysis of the distance covered per mice in an open field (**E**). Results are expressed as mean ± SEM per time-point in untreated (blue) and GW2580-treated groups (green). Statistics: two-way repeated-measure analysis of variance, followed by Bonferroni post-hoc test. Number of mice: 14 mice were used per experiment, per condition, and per time-point.

**Figure 2 brainsci-11-01643-f002:**
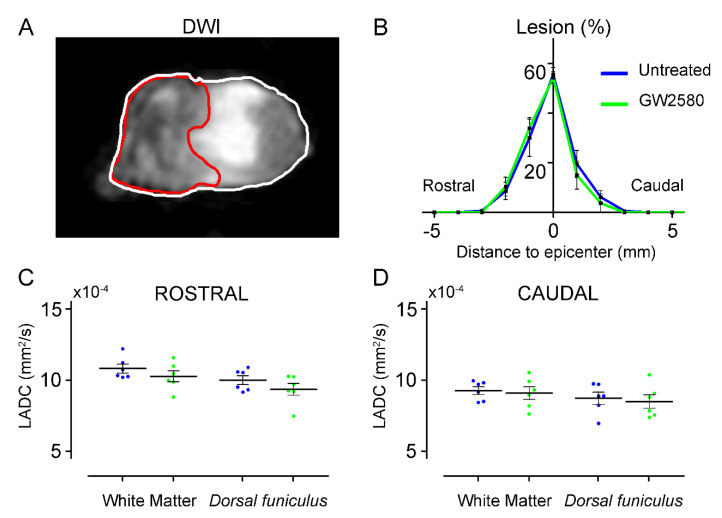
Long-term GW2580 treatment has no effect on lesion extension and longitudinal diffusivities. Representative ex vivo DW-MRI located at the lesion epicenter 6 weeks after injury (**A**). Axial spinal cord slice is delineated in white and lesion (here at the epicenter) in red. Line curves showing lesion percentages at epicenter, lesion extensions, and lesion volumes (area under the curve) in untreated (blue) and GW2580-treated (green) groups (**B**). Longitudinal apparent coefficient diffusion (LADC) rostral (**C**) and caudal (**D**) to the lesion in the white matter (excluding the *dorsal funiculus*) and the *dorsal funiculus*. Results are expressed as mean ± SEM per time-point in untreated (blue) and GW2580-treated groups (green). Each dot corresponds to a minimum of 8 sections (1 mm interval between each section). Statistics: Student’s unpaired *t*-test. Number of animals: 6 mice were used per experiment and per condition.

**Figure 3 brainsci-11-01643-f003:**
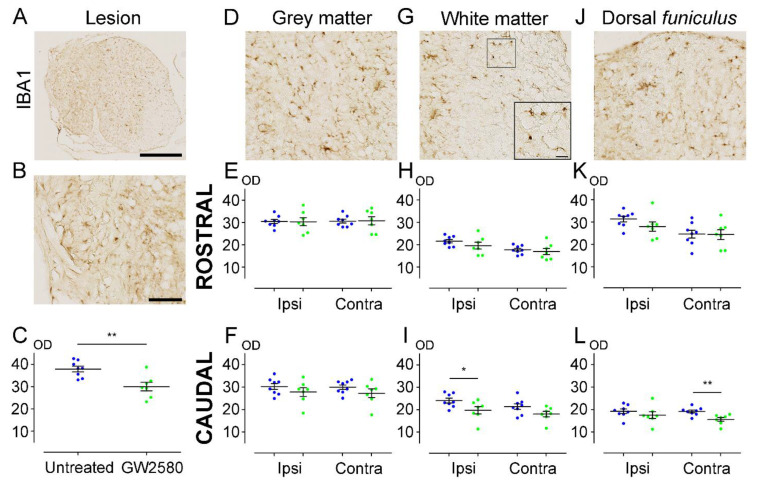
Long-term GW2580 treatment after SCI reduces IBA1 immunoreactivity in the injured spinal cord. Representative micrograph of IBA1 immuno-staining at the lesion epicenter (**A**). Zoomed-in micrographs in the lesion site (**B**), the grey matter (**D**), the white matter (**G**), and the *dorsal funiculus* (**J**) at the lesion epicenter. Six weeks after SCI, quantifications of IBA1 immunoreactivity were done in the lesion site (**C**), the grey matter (**E**,**F**), the white matter (**H**,**I**), and the *dorsal funiculus* (**K**,**L**). Optical densities were measured rostral (**E**,**H**,**K**) and caudal to the lesion site (**F**,**I**,**L**), ipsilateral, and contralateral to the lesion site. Results are expressed as mean ± SEM in untreated (blue) and GW2580-treated groups (green). Each dot corresponds to a minimum of 9 sections (630 µm interval between each section) analyzed per animal. Statistics: Student’s unpaired *t*-test. Number of mice: 8 for the untreated group and 7 for the GW2580-treated group. * *p* < 0.05; ** *p* < 0.01. scale bars: 500 µm (**A**); 100 µm (**B**,**D**,**G**,**J**); 20 µm (**G**) ( right bottom corner).

**Figure 4 brainsci-11-01643-f004:**
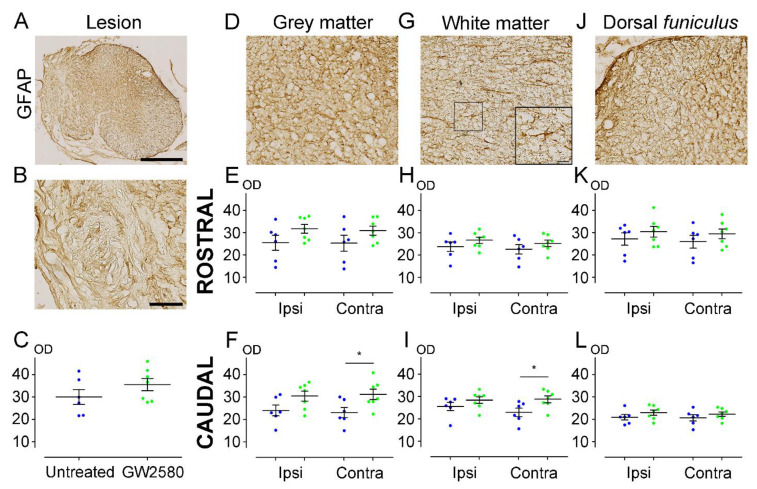
Long-term GW2580 treatment after SCI increases GFAP immunoreactivity in the spinal cord. Representative micrograph of GFAP immuno-staining at the lesion epicenter (**A**). Zoomed-in micrographs in the lesion site (**B**), the grey matter (**D**), the white matter (**G**), and the *dorsal funiculus* (**J**) at the lesion epicenter. Six weeks after SCI, quantifications of GFAP immunoreactivity were done in the lesion site (**C**), the grey matter (**E**,**F**), the white matter (**H**,**I**), and the *dorsal funiculus* (**K**,**L**). Optical densities were measured rostral (**E**,**H**,**K**) and caudal to the lesion site (**F**,**I**,**L**) ipsilateral and contralateral to the lesion site. Results are expressed as mean ± SEM in untreated (blue) and GW2580-treated groups (green). Each dot corresponds to a minimum of 9 sections (630 µm interval between each section). Statistics: Student’s unpaired *t*-test. Number of mice: 6 for the untreated group and 7 for the GW2580-treated group. * *p* < 0.05. scale bars: 500 µm (**A**); 100 µm (**B**,**D**,**G**,**J**). 20 µm (**G**) (right bottom corner).

**Figure 5 brainsci-11-01643-f005:**
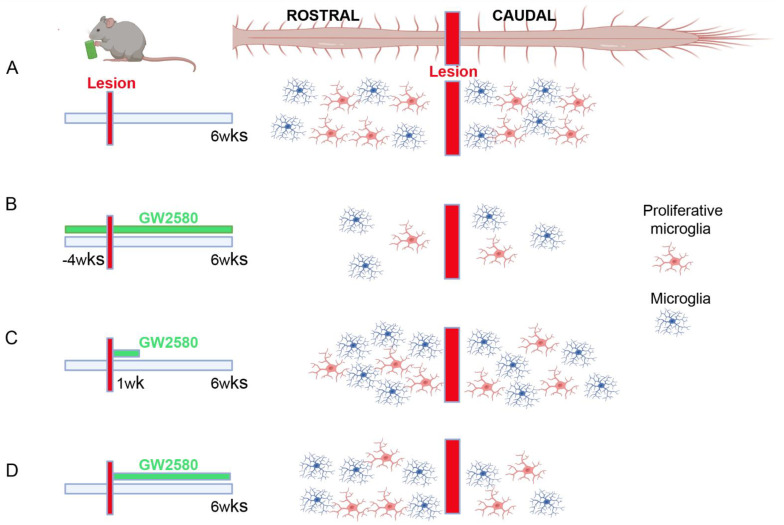
Microglia spinal distribution after injury depends on GW2580 treatment duration. Schematic drawings of the rostro–caudal distribution of microglia in the spinal cord 6 weeks after SCI. Microglia number and states are affected differently depending on the duration of GW2580 oral treatment. Homeostatic (blue) and proliferative (red) microglia distribution in the spinal cord after SCI without GW2580 treatment (**A**), after long-term pre- and post-SCI treatment (**B**), following short-term post-SCI treatment (**C**), and after long-term post-injury treatment (**D**).

## Data Availability

All data analyzed during this study are included in the published article and are available from the corresponding author on reasonable request.

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
