# Peer review of "Unlike Brief Inhibition of Microglia Proliferation after Spinal Cord Injury, Long-Term Treatment Does Not Improve Motor Recovery"

_brainsci, 2021, doi:10.3390/brainsci11121643_

Round 1
Reviewer 1 Report
In this study Poulen and colleagues assessed the effect of a 6-weeks GW2580 oral treatment after SCI on functional recovery and tissue reorganization. For this purpose, they treated SCI mice with a 6-weeks GW2580 oral treatment.
They observed that a long-term treatment with GW2580 after SCI, although effective in decreasing microglia proliferation, it did not improve motor recovery and had no effect on tissue reorganization as revealed by ex vivo diffusion-weighted magnetic resonance imaging and by the increasing astrocytic reactivity. They conclude that increasing the duration of GW2580-treatment is not beneficial for motor recovery after SCI.
The manuscript is potentially interesting, in particular from a translational point of view. There are several issues that the authors should address before considering their work mature for publication.
In particular, there was no effort to study the functional state of microglia. Although the microglia response after SCI is characterized by an early proliferation followed by a concomitant upregulation of pro- and anti-inflammatory factors, they focused exclusively on proliferation, neglecting the functional state, that could affect both functional recovery as well as tissue reorganization. How does a 6-week GW2580 oral treatment affect the microglia morphology and reactivity (see the recent paper of Catale et al., 2021 doi: 10.1016/j.bbi.2021.02.032).
Furthermore, in my opinion an aspect important that emerges from the study and that require to be further investigated is the increasing astrocytic reactivity, that, on the other hand, could strongly affect motor recovery and tissue reorganization.
Have the authors checked the expression of inflammatory markers in order to assess the astrocyte inflammatory response?
This because reactive astrocytes can exacerbate neurotoxicity and propagate the immune response by inducing infiltration of peripheral immune cells. Therefore, it is important to assess the effects of CSF1R inhibition with GW2580 on astrocyte reactivity after SCI.
Statistics
Considering that you have 1 dependent variable with more than one observation (at different weeks), the most appropriate static test for Catwalk experiments is a repeated measures ANOVA.
Minor issues:
Line 64: objective not objectiv.
Line 253: control not contrl.
Results:
Paragraph 3.3
Microglia reactivity…. The title is not appropriate considering that they checked only microglia proliferation.
Please in the figs. 3A and 4A insert a higher magnification in order to identify the specific glial population. In the same figs the scale bars are missing.
Author Response
We are very grateful to the reviewer’s comments and we have addressed all her/his comments. Moreover, a native English speaker has edited the revised version of our manuscript. Modifications as compared to the previous version can be followed since they are highlighted in green in our revised manuscript.
The manuscript is potentially interesting, in particular from a translational point of view. There are several issues that the authors should address.
1 - There was no effort to study the functional state of microglia. Although the microglia response after SCI is characterized by an early proliferation followed by a concomitant upregulation of pro- and anti-inflammatory factors, they focused exclusively on proliferation, neglecting the functional state, that could affect both functional recovery as well as tissue reorganization. How does a 6-week GW2580 oral treatment affect the microglia morphology and reactivity (see Catale et al., 2021 doi:10.1016/j.bbi.2021.02.032).
We agree with this reviewer that a complete characterization of the functional state of microglia after inhibiting their proliferation is of utmost importance. We have previously analyzed microglial molecular response following a short-term GW2580-treatment since it induces beneficial outcomes. We have now discussed these findings in the revised discussion section.
The morphological characterization of microglia is indeed of great importance, and we are including this approach in the research line that we are currently developing to better understand the beneficial outcome induced by a short-term GW2580-treatment. We hope that this reviewer will agree that is beyond the scope of the current manuscript since prolonged GW2580-treatment does not improve motor recovery and has thus limited (if any) translational interests.
We have now discussed the recent paper of Catale et al. Our results are totally along the same line since Catale et al. also point towards the importance of the time window of GW2580 exposure.
2 - Furthermore, in my opinion an aspect important that emerges from the study and that require to be further investigated is the increasing astrocytic reactivity, that, on the other hand, could strongly affect motor recovery and tissue reorganization.
Again, we fully agree with this reviewer that deciphering the crosstalk between microglia and astrocytes is of utmost importance in translational research. We have now better pointed that in the discussion.
We hope that this reviewer will also agree that is beyond the scope of the current manuscript. We are now developing a full research line to decipher the molecular interplay between microglia and astrocytes after short-term inhibition of microglia proliferation since it induces beneficial outcome in rodents and non-human primates following SCI.
3 - Have the authors checked the expression of inflammatory markers in order to assess the astrocyte inflammatory response? This because reactive astrocytes can exacerbate neurotoxicity and propagate the immune response by inducing infiltration of peripheral immune cells. Therefore, it is important to assess the effects of CSF1R inhibition with GW2580 on astrocyte reactivity after SCI.
We have not checked expression of inflammatory markers after a long-term GW2580-treatment. However, as part of our ongoing research we will carry out cell specific transcriptomic analysis following a 1-week treatment
4- Statistics
Considering that you have 1 dependent variable with more than one observation (at different weeks), the most appropriate static test for Catwalk experiments is a repeated measures ANOVA.
The test that had been done was indeed repeated measures ANOVA. It is now indicated in the Material and Methods section.
5- Minor issues:
Line 64: objective not objective, Line 253: control not contrl.
Errors had been corrected.
6 - Results:
Paragraph 3.3
Microglia reactivity…. The title is not appropriate considering that they checked only microglia proliferation.
We checked IBA1 expression, not microglia proliferation we have thus change the sub-title to “IBA1-microglial expression is decreased 6 weeks after SCI in the GW2580-treated mice”
7 - Please in the figs. 3A and 4A insert a higher magnification in order to identify the specific glial population. In the same figs the scale bars are missing.
We have now added higher magnifications in Figures 3 and 4. Scale bars are included in the figures and figure legends.
Reviewer 2 Report
The authors have used an inhibitor of microglial proliferation to test the effects of long term dosing on SCI model. Rather the matter conveyed are appropriate and relevant to the field. There are a concern
- The lack of uninjured control mice in every experiment reduces the rigor of the experiment, where the comparison of the disease model is not possible. Only when the injury has caused a significant impact on motor activity, the recovery effects of the drug can be discussed. In the absence of an uninjured animal group at every time point in Fig 1, the analysis is biased.
- The discussion can include a comparison of short term treatment (1 week) doi:10.7150/thno.61833 and other CSFR1 inhibitor https://doi.org/10.3389/fncel.2018.00368 and bring out the effectiveness of inhibiting microglia in improving the pathological condition.
Author Response
We are grateful to the reviewer’s comments and we have addressed all her/his comments. Moreover, a native English speaker has edited the revised version of our manuscript. Modifications as compared to the previous version can be followed since they are highlighted in green in our revised manuscript.
Rather the matter conveyed are appropriate and relevant to the field. There are a concern:
1 The lack of uninjured control mice in every experiment reduces the rigor of the experiment, where the comparison of the disease model is not possible. Only when the injury has caused a significant impact on motor activity, the recovery effects of the drug can be discussed. In the absence of an uninjured animal group at every time point in Fig 1, the analysis is biased.
We have now presented the behavioral data of uninjured vs injured mice as Supplementary Figure 1. We have also updated the Material and Method and result sections accordingly. It attests that the injury causes a significant deficit on motor activity. For the CatWalk analysis, static and dynamic parameters strongly depend on the speed of the animal. Injured animals display a slower locomotion after spinal cord injury; it is thus difficult to directly compare the data between the injured and un-injured animals [1]. For this reason and for the clarity of the presentation, we have not presented these data in Figure 1 but rather as supplemental data in the revised version of our manuscript.
- Clarke, K.A. and J. Still, Gait analysis in the mouse. Physiol Behav, 1999. 66(5): p. 723-9.
2 The discussion can include a comparison of short-term treatment (1 week) doi:10.7150/thno.61833 and other CSFR1 inhibitor https://doi.org/10.3389/fncel.2018.00368 and bring out the effectiveness of inhibiting microglia in improving the pathological condition.
As suggested, we have now extended the first paragraph of the discussion and have included the comparisons with our two previous studies, as schematized in Figure 5.
Reviewer 3 Report
In this manuscript, the authors describe their results that increasing the duration of GW2580-treatment is not beneficial for motor recovery after SCI. After treatment with GW2580, the microglial reactivity would decrease and the astrocytic reactivity could increase. However, in their previous studies (ref9), similar results have been published. Although the author mentioned the ref in the discussion part, that is still not enough. It is not clear the difference between both studies. Both studies use the same model and treatment strategy.
The author thinks short-term treatment by GW2580 could improve motor recovery. The current results do not include any short-term treatment experiments. Based on the results, the author only makes a conclusion that long-term treatment by an inhibitor after SCI can not promote motor recovery. The results for short-term treatment are from their another paper (ref10). So “the short term…” is not fit for the manuscript.
The author found the expression of GFAP is increasing after treatment of GW2580. That will be interesting. GW2580 is an inhibitor of microglial proliferation, but no studies show if it can influence astrocytes after SCI. However, the authors do not fully explain the relationship between them.
Minor:
- There should not be italic in lines 157-175
Author Response
We are grateful to the reviewer’s comments and we have addressed all her/his comments. Moreover, a native English speaker has edited the revised version of our manuscript. Modifications as compared to the previous version can be followed since they are highlighted in green in our revised manuscript.
In this manuscript, the authors describe their results that increasing the duration of GW2580-treatment is not beneficial for motor recovery after SCI.
1 - Although the author mentioned the ref (“of their previous publication”) in the discussion part, that is still not enough. It is not clear the difference between both studies. Both studies use the same model and treatment strategy.
As suggested, we have now clarified and extended the first paragraph of the discussion, in order to better present treatment strategies and outcome as schematized in Figure 5. The Graphical Abstract also summarizes these data.
2 - The results for short-term treatment are from their another paper (ref10). So “the short term…” is not fit for the manuscript.
We agree with the reviewer and have revised the title accordingly as follow: “Unlike brief inhibition of microglia proliferation after spinal cord injury, long-term treatment does not improve motor recovery”
3 - The author found the expression of GFAP is increasing after treatment of GW2580. GW2580 is an inhibitor of microglial proliferation, but no studies show if it can influence astrocytes after SCI. However, the authors do not fully explain the relationship between them.
We fully agree with this reviewer that deciphering the crosstalk between microglia and astrocytes is of utmost importance in translational research. We have now better pointed this issue in the revised discussion.
We hope that this reviewer will agree that it is beyond the scope of the current manuscript. We are now developing a full research line to decipher the molecular interplay between microglia and astrocytes after short-term inhibition of microglia proliferation since it induces beneficial outcome not only in rodents but also non-human primates following SCI.
Minor:
1 - There should not be italic in lines 157-175
It had been corrected.
Round 2
Reviewer 1 Report
The effort made by Authors to improve the paper has been minimal.
I have asked for a higher magnification of Iba-1 and GFAP staining in the figs. 3A and 4A, but the zoom pictures (Zoomed-in micrographs in the lesion site) presented do not allow to identify the two glial populations. Please insert a picture (at very high manification) of the IHC staining in which is clearly visible a microglia and an astrocyte with their own specific features. It is impossible to recognize the specific cell population. The Dab reaction, especially for GFAP, is quite saturated, so in this case the OD it not suitable.
Seeing the high magnification I doubt that the OD could be the best measure of cell proliferation. In my opinion, the count of the number of the cells (microglia and astrocytes) would be more suitable as index of proliferation than OD.
In Fig. 3A and 4A at low magnification please put some coloured boxes in order to indicate exactly the area shown in panels D-G-J.
Check Typos (Transcriptomic not transcriptmic line 362...).
Author Response
We are delighted to address all reviewer comments. Modifications as compared to the previous revised version can be followed since they are highlighted in yellow in our newly revised manuscript
1- The effort made by Authors to improve the paper has been minimal.
We felt that we had address all comments of the 3 reviewers, particularly considering we were given only 10-days.
2 - I have asked for a higher magnification of Iba-1 and GFAP staining in the figs. 3A and 4A, but the zoom pictures (Zoomed-in micrographs in the lesion site) presented do not allow to identify the two glial populations.
Within the lesion epicenter, there are only few, if any GFAP, positive cells; this is the hallmark of spinal cord injury. Indeed, astrocytes form a scar that surrounds the lesion epicenter. However, since we are using hemisection model of the spinal cord injury, the contralateral side of the lesion present astrocytes.
Therefore, we have measured OD separately in the rostral, caudal, and lesion epicenter. Our images in Figs. 3B and 4B highlight tissue disorganization at the lesion epicenter. Nevertheless, as insisted by the reviewer, we are now including higher magnification images in Figs. 3G and 4G to illustrate the differences in morphology between different microglia populations (ramified and amoeboid) and astrocytes (satellite cells) in the white matter of the spinal cord. We have chosen the white matter of the spinal cord due to clear appearance of glial morphology.
3 - Please insert a picture (at very high manification) of the IHC staining in which is clearly visible a microglia and an astrocyte with their own specific features. It is impossible to recognize the specific cell population. The Dab reaction, especially for GFAP, is quite saturated, so in this case the OD it not suitable.
As mentioned above, we have now inserted higher magnification images of IBA1 (Fig. 3G), and GFAP (Fig. 4G). We agree that DAB staining is saturated at the lesion epicenter, however it is not saturated rostral and caudal to the lesion. We are analyzing over 1.2 cm segment of the spinal cord using serial sections covering 0.6mm rostral to the lesion, lesion epicenter, and 0.6mm caudal to the lesion on the same slide. We thus selected the optimum DAB staining for rostral and caudal segments. In addition, for a given antibody, all sections were stained in parallel to avoid differences in DAB timing between experiments. Moreover, all slides for a given antibody were scanned in parallel using NanoZoomer® (NanoZoomer Digital Pathology (NDP) System and NDP view software, Hamamatsu City, Japan). This is an automated apparatus that uses constant exposure time and light intensity for all slides and sections. Furthermore, all images for a given antibody were exported using NDPview® software with the same exporting parameters. Finally, all quantifications were done blindly by allocating a specific code for given slides and the code were not revealed until all quantifications were completed. For all above reasons, we thus think that OD measurement is suitable to measure intensity of a given signal i.e., GFAP and IBA1 expression between different groups.
4 - Seeing the high magnification I doubt that the OD could be the best measure of cell proliferation. In my opinion, the count of the number of the cells (microglia and astrocytes) would be more suitable as index of proliferation than OD.
Indeed, we agree that OD is not a measure of cell proliferation. At the end of the treatment we are analyzing the differences in glial reactivity rather than proliferation.
5 - In Fig. 3A and 4A at low magnification please put some coloured boxes in order to indicate exactly the area shown in panels D-G-J.
Adding 4 boxes in the lower mag images would render the overview of damaged tissue in the lesion epicenter difficult to assess. However, as suggested by the reviewer, we have added a box to identify the glia morphology and location of the zoom in IBA1 (Fig. 3G) and GFAP (Fig. 4G).
Check Typos (Transcriptomic not transcriptmic line 362...).
The error had been corrected
Reviewer 2 Report
Accept in the current format
Author Response
Thank you for your review.
Reviewer 3 Report
The authors have done a lot of job for the questions and suggestions that I raised in my initial review. However, there are still some questions that should be considered.
1. Although the authors change the title of the manuscript and remove “Short-term”, the conclusion is still focused on short time (line30-31 and line 429-431). The question is still not resolved. The author did not do short-term treatment in the current study. I think we cannot make a conclusion for short-term treatment.
2. In result 1, no results show microglia proliferation has been inhibited. Maybe, the title should be changed.
3. In the study, after being treated by GW2580, the proliferation of microglia is decreased and reactive astrocytes are increasing. Reactive astrocytes still have benefits on wound healing, tissue remodeling, and motor functions (review paper PMID: 29054466, 14999065, 28737515). The author should discuss this.
Minor:
1. I think “regulation” in line 53 could be removed.
2. In “Supplementary Figure 1”, the statistic should be done by time points.
3. The sentence in lines 338-340 is confusing.
4. The sentence in lines 391-393 is confusing.
Author Response
We have now addressed all reviewer’s comments. Modifications as compared to the previous revised version can be followed since they are highlighted in yellow in our revised manuscript.
The authors have done a lot of job for the questions and suggestions that I raised in my initial review. However, there are still some questions that should be considered.
- Although the authors change the title of the manuscript and remove “Short-term”, the conclusion is still focused on short time (line30-31 and line 429-431). The question is still not resolved. The author did not do short-term treatment in the current study. I think we cannot make a conclusion for short-term treatment.
We have now removed the last sentence referring to the short-term treatment in the abstract. We have added reference to the last part of the discussion that mentioned our previous publication focusing on short time treatment after SCI.
- In result 1, no results show microglia proliferation has been inhibited. Maybe, the title should be changed.
As suggested, we have now revised the title in the result section as follows: “Long-term GW2580-treatment after SCI does not improve motor recovery”
- In the study, after being treated by GW2580, the proliferation of microglia is decreased and reactive astrocytes are increasing. Reactive astrocytes still have benefits on wound healing, tissue remodeling, and motor functions (review paper PMID: 29054466, 14999065, 28737515). The author should discuss this.
We have now included this reference in the Discussion section as follow: “Moreover, it had been shown that astrocytes change their phenotype depending on the environment (for review see [(Okada, Hara et al. 2018)]), thus a 6-weeks GW2580-treatment may induce a different response of activated astrocytes than a 1-week treatment duration that eventually led to an absence of functional recovery”.
Minor:
- I think “regulation” in line 53 could be removed.
We have removed “regulation”.
- In “Supplementary Figure 1”, the statistic should be done by time points.
As suggested, we have now included Bonferroni tests by time points in Supplementary Figure 1
- The sentence in lines 338-340 is confusing.
We have modified the sentence as follow: “Following a 1-week GW2580-treatment starting immediately after SCI in mice, we observed an increased microgliosis six weeks post-lesion (i.e. 5 weeks after completion of the treatment, Figure 5C)”.
- The sentence in lines 391-393 is confusing.
We have modified the sentence as follow: “Conversely, a 1-week GW2580-treatment led to an increase in microglial reactivity 6-weeks after SCI compared to untreated mice (Figure 5C) although astrocyte reactivity was similar in both groups (personal communication).
Reference
Okada, S., M. Hara, K. Kobayakawa, Y. Matsumoto and Y. Nakashima (2018). "Astrocyte reactivity and astrogliosis after spinal cord injury." Neurosci Res 126: 39-43.